# Investigation of Global Trends of Pollutants in Marine Ecosystems around Barrang Caddi Island, Spermonde Archipelago Cluster: An Ecological Approach

**DOI:** 10.3390/toxics10060301

**Published:** 2022-06-01

**Authors:** Ismail Marzuki, Early Septiningsih, Ernawati Syahruddin Kaseng, Herlinah Herlinah, Andi Sahrijanna, Sahabuddin Sahabuddin, Ruzkiah Asaf, Admi Athirah, Bambang Heri Isnawan, Gatot Supangkat Samidjo, Faizal Rumagia, Emmy Hamidah, Idum Satia Santi, Khairun Nisaa

**Affiliations:** 1Department of Chemical Engineering, Fajar University, Makassar 90231, South Sulawesi, Indonesia; 2Research Institute for Coastal Aquaculture and Fisheries Extension, Maros 90512, South Sulawesi, Indonesia; earlyseptiningsih@gmail.com (E.S.); ernawatisyahruddin71@gmail.com (E.S.K.); hjompa@yahoo.com (H.H.); idsarsompa@gmail.com (A.S.); s.abud_din@yahoo.co.id (S.S.); qiaasaf@gmail.com (R.A.); m.athirah@gmail.com (A.A.); 3Department of Agrotechnology, Universitas Muhammadiyah Yogyakarta, Bantul 55183, DI Yogyakarta, Indonesia; bambanghi@umy.ac.id (B.H.I.); supangkat@umy.ac.id (G.S.S.); 4Study Program of Fisheries Resource Utilization, Faculty of Fisheries and Marine, Khairun University, Ternate 97719, North Maluku, Indonesia; faizal.rumagia@unkhair.ac.id; 5Department of Agrotechnology, Universitas Islam Darul ‘Ulum, Lamongan 62253, Jawa Timur, Indonesia; emmyhamidah@unisda.ac.id; 6Department of Agrotechnology, Institut Pertanian Stiper, Yogyakarta 55283, DI Yogyakarta, Indonesia; idum@instiperjogja.ac.id; 7National Research and Innovation Agency (BRIN), Jakarta 10340, DKI, Indonesia; nisauicha27@gmail.com

**Keywords:** pollutants, microplastics, heavy metals, polycyclic aromatic hydrocarbons, pesticide residues, medical waste

## Abstract

High-quality marine ecosystems are free from global trending pollutants’ (GTP) contaminants. Accuracy and caution are needed during the exploitation of marine resources during marine tourism to prevent future ecological hazards that cause chain effects on aquatic ecosystems and humans. This article identifies exposure to GTP: microplastic (MP); polycyclic aromatic hydrocarbons (PAH); pesticide residue (PR); heavy metal (HM); and medical waste (MW), in marine ecosystems in the marine tourism area (MTA) area and Barrang Caddi Island (BCI) waters. A combination of qualitative and quantitative analysis methods were used with analytical instruments and mathematical formulas. The search results show the average total abundance of MPs in seawater (5.47 units/m^3^) and fish samples (7.03 units/m^3^), as well as in the sediment and sponge samples (8.18 units/m^3^) and (8.32 units/m^3^). Based on an analysis of the polymer structure, it was identified that the dominant light group was MPs: polyethylene (PE); polypropylene (PP); polystyrene (PS); followed by polyamide-nylon (PA); and polycarbonate (PC). Several PAH pollutants were identified in the samples. In particular, naphthalene (NL) types were the most common pollutants in all of the samples, followed by pyrene (PN), and azulene (AZ). Pb^+2^ and Cu^+2^ pollutants around BCI were successfully calculated, showing average concentrations in seawater of 0.164 ± 0.0002 mg/L and 0.293 ± 0.0007 mg/L, respectively, while in fish, the concentrations were 1.811 ± 0.0002 µg/g and 4.372 ± 0.0003 µg/g, respectively. Based on these findings, the BCI area is not recommended as a marine tourism destination.

## 1. Introduction

Marine ecosystems are formed by two main components: biotic (organic) and abiotic (inorganic) elements [1,2,3]. The seascape is vast and is influenced by many factors, causing the characteristics of the sea area to vary [4,5]. Each marine area has different characteristics and ecosystems, such as deep-sea, poles, beaches, coasts, coral reefs, and tides [6]. The types and amounts of materials in marine ecosystems are not limited, and comprise both those that are naturally occurring and those that are produced due to human activities and dynamics [7,8]. Most human waste material can occur in all of the regions of the earth. Due to the influences of ecology and gravity, garbage has the potential to end up in the sea, either intentionally or due to negligence in managing the marine environment [9,10,11].

Human waste materials that enter marine waters are classified as marine debris and generally contain hazardous and toxic components, such as heavy metals (HM), polycyclic aromatic hydrocarbon components (PAH), microplastics (MP), pesticide residues (PR), medical waste (MW), and radioactive waste. They can be categorized as global trending pollutants (GTP) [12,13,14,15,16,17,18,19,20]. The term global trending pollutants was introduced to the world community to describe at least five types of pollutants (MP, PAH, PR, MW, and HM), which are pollutant materials that have become global environmental issues to date, especially for marine ecosystems [14,18,21,22,23,24].

The handling of GTP as toxic components is a problem that has occurred for a long time, with impacts including contributions to the triggering of global warming and climate change; thus, they have become a severe problem for many countries in terms of handling and managing them [25,26,27,28]. Many countries have difficulty handling these GTP, especially developing countries with relatively low economic levels. The concern of the world’s population about the massive quantity of GTP is indicated by the issuance of G20 recommendations for developed countries to reduce global carbon production [2,29]. Whether we realize it or not, these global trending pollutants’ inputs enter the ocean continuously and are difficult to prevent, and it is feared that the toxic components in the waste have the potential to cause anthropogenic hazards in the oceans [3,5,30].

The interactions among materials in the sea occur continuously and produce various types of new compounds, due to the natural ability of the sea to maintain the sustainability of life in the ocean [31,32]. Photodegradation due to sunlight, interaction by ocean currents, mobilization of materials due to differences in the waters, effects of salinity, consequences arising from temperature differences, and collisions between materials due to tidal movements in coastal areas are natural processes of the sea that occur continuously [17]. The occurrence of these events in marine ecosystems enriches the production of components with diverse marine biological and non-biological materials [33,34,35,36,37].

The effect of applying polymer technology to the production of various types of industrial equipment, household appliances, and plastic-based packaging was previously predicted to produce components that are harmful to human health and the environment, such as microplastics. However, it cannot be avoided due to the rapid flow of human needs [18,19,38,39]. Pesticide residue pollutants, the birth of synthetic products for activities to accelerate and increase agricultural production, and fulfilment of the need for treatment with various types of synthetic drugs and medical equipment were studied before. These factors have the potential to cause negative impacts, but cannot be suppressed due to the swift pace of demand [40,41,42]. Estimates of the emergence of PAH-type components in petroleum processing and fossil combustion, heavy metal contaminants in mineral processing, and the introduction of various metal-based equipment were all previously predicted. However, this cannot counter the solid demand of meeting human needs, particularly in terms of preserving life [43,44].

The distribution of GTP is almost even among many countries, and their presence is especially significant in marine and coastal areas. GTP have toxic and carcinogenic properties, are generally difficult to decompose, have accumulative properties, and can combine and even become assimilated into the processes of the objects, especially their metabolic processes [45,46]. The particle size of MP is minimal (˂5 nm), allowing it to contribute to the cell division of living things, not least in marine ecosystems, such as fish and other marine biota and various marine biological components, which, at the end of the biogeochemical cycle, have the potential to become materials that accumulate in human foodstuffs [47,48].

GTP are pollutants that are very difficult to decompose through biodegradation, bio-adsorption, and bioreduction methods, including destruction methods. The treatment approach using bioremediation methods for global trending pollutants is challenging to conduct on a large scale with a technological approach [49,50]. Bioremediation microorganisms can be used, where the natural aid of biological materials, such as bio-adsorbent plants, is employed. However, they certainly cannot balance the speed of increase in the volume of pollutants entering the environment, when operating at a limited rate, and especially in the most vulnerable marine ecosystems [51]. This level of pollution could become a severe threat to the Earth’s population, especially in marine areas and ecosystems, which could act as giant containers for almost all types of waste that are left over from human activities and dynamic natural processes [52,53].

The phenomenon of tourism activities that make marine ecosystem objects into marine tourism areas is occurring in many countries, including Indonesia [54]. Exploiting marine areas as tourist areas by presenting a variety of marine beauty characteristics, both on the surface and deep in the sea, is tempting because many people are interested in them [55]. Barrang Caddi Island (BCI) is one of the five islands included in the Marine Tourism Area (MTA), in addition to Samalona Island, Kodingareng Keke, Barrang Lompo, and Langkai. These islands are designated as an MTA that are managed by the local government of Makassar City, South Sulawesi [56]. The islands in the MTA, especially BCI, are included in a national marine conservation park and are part of the Spermonde Archipelago Cluster, which is world famous among marine sponge observers, because it is home to tens of thousands of species and marine sponge populations. The Spermonde Archipelago Cluster is known worldwide as a sponge study and research laboratory [57,58].

Barrang Caddi Island (BCI) in the south is directly adjacent to Makassar City. Along the coastline, there are the Soekarno Hatta seaports and fish landing ports. This coastal area operates two regional and two industrial hospitals and several large and medium sized hotels [54,55]. The beach is also full of marine tourism objects, which are visited by many domestic and foreign tourists. All forms of operations and services around BCI have the potential to produce waste containing GTP group contaminants. This contradicts the island’s status as a marine tourism area and the viability of the various types of sponge living around it [59,60].

Research in this area is exciting and essential. We present an update on the data and information related to GTP contaminants in a review of the seawater, sediments, fish, and sponges, as well as the dual role of BCI as part of an MTA and the living area of various types of sponge [14,15,16,17]. The data presented are expected to act as a reference for the management of BCI as part of the MTA to prevent marine tourism activities from adding to the burden of island sustainability for marine life by ensuring that the balance of marine ecosystems is maintained and, most importantly, that the sustainability of the sponge population is maintained.

## 2. Materials and Methods

### 2.1. Materials and Equipment

Sediment samples, seawater, sponges, and demersal fish were the main materials. Other materials included H_2_O_2_ p.a., 30%, Fe(II) 0.05 M, N-hexane p.a., ethanol p.a., standard analysis of PAH (anthracene) 1000 mg/L (Supelco^®^), Na_2_SO_4_, H_2_SO_4_ p.a., NaCl p.a., HCl p.a., NaOH p.a., HNO_3_ p.a., HClO_4_ p.a., Pb(NO_3_)_2_ p.a.(standard solution of Pb analysis 1.000 mg/L), and CuSO_4_ (standard solution of Cu 1000 mg/L), which were all obtained from Sigma-Aldrich, Saint Louis, MO, USA. Equipment used included the Fourier Transform Infra-red (FTIR) Shimadzu IR Prestige-21, Shimadzu Europa GmbH, Duisburg, Germany, a Gas Chromatography/Mass Spectrometer (GC/MS) from Agilent Technologies 7890A, Santa Clara, CA, USA (the operating conditions for GC/MS max. temperature 350 °C, the increase in temperature of 10 °C, every 5 min, pressure 18.406 psi), a Helium gas carrier, a speed of 150 mL/min, capillary column (Agilent 19019S-436HP-5 ms, Santa Clara, CA, USA), (dimensions of 60 m × 250 μm × 0.25 μm, the pressure 18,406 psi, separation 26,128 cm/sec, a retention time of max. 30 min) [12,44], an Atomic absorption spectroscope (AAS) (variant type AA240FS), a Muffle Furnace (type Thermolyne F6010, Shanghai, China), Portable Water (Quality AZ 8361, AZ Instrument Corp., Taichung, Taiwan), a set of glassware (Pyrex), a plankton net (mesh size 0.4 mm), stainless-steel mesh filters (5 mm and 0.3 mm), a sieve shaker, a density separator, and a vacuum pump [33,36,48].

### 2.2. Sampling

The total sample was 36 packages, each consisting of three sample packages of each type (sediment, seawater, sponges, and fish). The samples were obtained around the BCI waters at three different sampling points (Figure 1) [3,21,61,62,63]. Physical characteristics were observed at each station, such as the sampling coordinates, pH, salinity, electrical conductivity (EC), total dissolved solids (TDS), and others (Table 1). For the fish and sponge samples, a morphological analysis was also carried out to determine the species in each sample. Sampling was carried out following standard sampling methods. Each sample was put in a box in the laboratory for immediate preparation until it was ready to be analyzed using the appropriate instrument [64,65].

Sediment, sponge, seawater, and fish samples were obtained from three different sampling stations around BCI. This island is included in the Marine Tourism Area of Makassar City [1,5] The distance between BCI and the coast of Makassar City is ±7.5 km. The small red circles (Figure 1) are non-sampling areas, referred to as stations, which are coded ST 1, ST 2, and ST 3 [22,24].

According to the data (Table 1), the physical characteristics of the sampling stations are similar to those of other marine areas around them, especially in the Spermonde Islands cluster [24,33]. The data show that the seawater shows no physical signs that the area is exposed to the contaminants that fall into the GTP category.

### 2.3. Sample Preparation

Samples were prepared according to the type of analysis conducted, based on the target data. Sample preparation was carried out to determine the abundance of microplastics by filtering using a plankton net (adjusted for the samples of fish, sponges, and sediments). The samples were dissolved and the organic matter and the microplastic particles were separated [66]. The total abundance of microplastics was calculated in each sample [19,28]. The microplastic polymer structure was analyzed using microplastic particles collected based on differences in density, put in a cuvette, and then run using FTIR [67,68]. This instrument can produce a chromatogram as a spectrum in the form of a wavenumber that can be converted into the chemical structure with functional groups. Based on the functional groups, the type of MP can be predicted. Seven types of MP pollutants may be present in a sample, among which the most common are polypropylene, polyethylene, polystyrene, and polycarbonate [69,70].

Sample preparation for the analysis of the abundance and types of PAH was completed using an extraction ethanol, where the ethanol was used to extract all chemical components in the sample. Then extraction with N-hexane as a solvent was executed to separate the nonpolar components (PAH) [71]. The N-hexane extraction was run using GC/MS [72]. The visible chromatograms were analyzed to determine the abundance and types of PAH components present. Sample preparation for the analysis of the concentrations of Pb^+2^ and Cu^+2^ contaminants in the sample was carried out with the dry destruction method for the sediment, fish, and sponge samples, while the alkaline digestion method was used for seawater samples [73,74,75]. Each sample was dissolved with a concentrated acid solvent. Samples were injected into the AAS [44,45,55,76].

### 2.4. Sample Measurement

There were five types of pollutant in the GTP category. However, in the analysis of each sample, the identification of microplastics, PAH, and heavy metal contaminants was prioritized, while the identification of pesticide residues and medical waste was not prioritized to limit the publication’s scope. However, these two types of contaminants can be viewed, together with PAH, based on GC/MS chromatogram data:The analysis of microplastic components, especially the total abundance of microplastics was completed using 30% H_2_O_2_ solvent. Measurement of the total abundance of microplastics was carried out in triplicate. The polymer structure was analyzed as the main characteristic of each microplastic, using FTIR combined with GC/MS chromatogram data [26,76,77];The types and abundance of the PAH contaminants were analyzed based on GC/MS chromatogram data. Information on the pesticide residue contaminants and media waste can also be obtained from the GC/MS chromatogram [24,76,77]. However, the determination of components was more specific, because in general, pesticide residues and several types of medical waste have chemical structures with reactive groups that should be identified using a combination of pyrolysis-GC/MS [54,78,79];Determination of the concentration of heavy metal contaminants, particularly exposure to Lead (Pb) and Copper (Cu) ions, was conducted for each sample according to AAS absorption at the maximum wavelength (Pb: λmax. 228.9 nm and Cu: λmax. 324.7 nm) [33,34,80]. The method for determining the pollutant concentrations of Pb^+2^ and Cu^+2^ first makes a calibration curve to make ten series of Pb^+2^ and Cu^+2^ standard solutions whose estimated sample concentrations fall within the range of the standard solution, then the absorption of each concentration is measured. Then, the standard deviation and slope determination calculations are carried out. The concentration of each sample (Pb^+2^ and Cu^+2^) was calculated based on the absorption obtained from AAS after being plotted into the regression equation [22,33,80]. Determination of pollutant concentrations of Pb^+2^ and Cu^+2^ carried out measurements of three replications for each type of sample obtained at three different sampling points. The data from the measurement results were calculated on average and summarized in a table. The quality of seawater according to the quality standard for pollutants Pb^+2^ and Cu^+2^ is a maximum of 0.05 mg/L. The quality standard for fish and other non-spongy biota is a maximum of 0.008 mg/L [55,76,80,81,82]. The maximum limit is not specified for sponges because they are included in the category of biota that are not eaten. The maximum limit of Pb^+2^ and Cu^+2^ for sediment is 0.10 µg/g. The quality of seawater and fish in BCI is determined by comparing the average pollutant concentration calculated compared to the standard for seawater and fish [4,10,59,83].

### 2.5. Data Presentation and Analysis

The presentation of analytical data as quantitative data was completed, using the appropriate equation. Data on the total abundance of MP, Pb, and Cu ion contaminant concentrations are displayed. The qualitative analysis data are presented based on the chromatogram results [26,28,84]. The polymer structure types of microplastic contaminants were analyzed using the FTIR chromatogram reading data, while data on PAH exposure types, including the pesticide residues and medical waste contaminants in each sample, were analyzed using GC/MS chromatography [85,86,87].

## 3. Results

Five types of pollutant fall into the GTP group, and the target of the GTP contaminant analysis was samples of sediment, sponge, seawater, and fish obtained from BCI (Makassar City MTA). The priority for the analysis was MP-type pollutants, of which the main hydrocarbon components are PAHs and heavy metal pollutants (HM). Pesticide residue pollutants and medical waste are not detailed, because of concerns that the data obtained are not significant due to the limitations of the analytical instrument as, ideally, a combined pyrolysis-GC/MS analysis tool should be used to analyze such substances [86,87].

### 3.1. Microplastic Pollutant Analysis

A specific morphological analysis of fish and sponge samples is essential because each type of marine biota has a different lifestyle and habitat, so the response to GTP contaminants or contaminant species (MP, PAH, PR, MW, and HM) can vary. The results of the morphological analysis of fish and sponge samples obtained at each station are presented in full in Table 2.

The grouping of samples by pairing sediment and sponge, and seawater and fish to present the analysis results was based on the similarity of the media. The growth media and the habitat of sponges are more dominant in sediment, while the habitat and growth of fish are highly dependent on water. Under these conditions, the type of MP pollutant in fish bodies is estimated to be similar to the types of MP in water, in the bodies of sponges, and in sediments [21,22].

Two parameters were used to analyze the MP pollution in the sample: the total MP abundance (Figure 2 and Figure 3), and the MP type (Figure 4 and Figure 5).

The total MP abundance in the sediment samples was similar to the total MP abundance in the sponge samples (Figure 2). These results illustrate that MP exposure in sediment samples is relatively consistent with that in the sponge samples and is generally thought to be dominated by fiber and fragment types [18,26]. This is based on the difference in density between the two types of MP, which have higher specific gravity than water and film-type MP, so the two types of MP accumulate in greater concentrations on the bottom of the water [27,64]. The relatively settled lifestyle of sponges and their method of obtaining food nutrients by sucking particles from the mud and removing the dregs (filter feeder) are solid arguments for why the total abundance of MP in sediments is similar to that in sponges [26,27,39].

Total MP abundance in fish samples was higher than the total MP abundance in seawater samples. This occurred at all three sampling points (Figure 3). This result is entirely rational, as fish mobility is wider, so the presence of fish is not expected to always be in the zone of seawater exposed to MP [38,66]. The type of MP that fish samples are exposed to is also thought to be dominated by film-type MP, which have a lower density than water, meaning that they float on the surface zone, and are prone to being swallowed by fish, as the fish consider them to be food. The difference in the total MP abundance of fish samples between stations 1, 2, and 3 (Figure 3) is thought to be influenced by the different fish samples at each station (Table 2). The average total abundance of MP in the samples was 8.18 units/m^3^ in sediment, 8.32 units/m^3^ in sponge, 5.47 units/m^3^ in seawater, and 7.03 units/m^3^ in fish [19,22,28].

The type of MP pollutant was based on the polymer structure thought to be present in each analyzed sample. This was determined by analyzing the spectrum read by the FTIR chromatogram. The FTIR spectrum (Figure 4 and Figure 5) shown for each type of sample is that obtained from station 1 (ST 1), which was considered representative of the samples collected at ST 2 and ST 3 [28,39].

The analysis of the MP pollutants was completed with an FTIR chromatogram and was based on the unique monomer structures of functional groups in each type of MP. Seven types of MP are commonly found in marine ecosystems [18,70]. Their respective monomers are as follows: polypropylene (PP) with the molecular formula (C_3_H_6_)n; polyethylene (PE), a monomer of ethylene (C_2_H_4_)n; polystyrene (PS) (C_8_H_8_)n; polycarbonate (PC), which has a carbonyl functional group (O-(C=O)-O)n; polyamide-nylon (PA), with the monomer formula NH_2_(CH_2_)_6_.NHCO(CH_2_)_4_COOH; polyvinyl chloride (PVC) with unique components, such as chloride (C_2_H_3_Cl)n; and acrylonitrile butadiene styrene (ABS), which is composed of three monomers with the formula (C_8_H_8_)x(C_4_H_6_)y(C_3_H_3_N)z [18,19,26,27,38,64]. The results of the MP-type analysis based on the FTIR spectrum are displayed for each sample obtained at ST 1 (Figure 4). The MP, PP, and PS types existed in all of the samples. The ABS type was found in sediment, sponge, and seawater samples, while PVC was identified only in sea-water samples [19,38].

The types of MP identified in the seawater and fish samples did not differ significantly from the types of MP identified in the sediment and sponge samples (Figure 5). All results for the identification of MP species, based on the polymer structures found in the samples obtained at the three stations (ST 1, 2 and 3) and shown on the FTIR chromatogram, are presented more fully in Table 3.

In terms of the types of MP identified in each sample at the sampling points (ST 1–3) (Table 3), the pollutant MP type PA was identified in all of the samples, except for the seawater samples (ST 1 and 2). The PC type was seen in most of the samples, except for the sponge and seawater samples (ST 1) [64,68]. The PE pollutants were also recorded in most of the samples, except for the seawater samples (ST 3). PP contaminants were not found in the sponge or fish samples (ST 3), while PS was found in all of the samples at each station. PVC-type pollutants were identified in all of the sample types obtained from ST 3, and ABS-type pollutants were identified in sediment, sponge, and seawater samples from ST 1 only [66,68].

### 3.2. PAH Pollutant Analysis

The identification of hydrocarbon pollutants, especially the PAH type, and other hydrocarbon components in the samples was completed using the GC/MS chromatogram data. The parameters used to determine the presence of PAH-type pollutants in the samples were based on several recorded chromatogram data units, namely, the peak number, retention time, peak height as a manifestation of the component abundance, the content in percentage units in relation to the total PAH, and the component name.

According to the source (ST 1–3), each sample’s complete details regarding the PAH-type hydrocarbon pollutants are presented in tabular form (Table 4, Table 5 and Table 6). As shown in Table 4, Table 5 and Table 6, the peak numbers are not sequential, and it is known that each peak is identical to one component [88,89]. Some peak numbers are not included in the table, due to several different reasons: first, the peak that appears is not a PAH component. Second, it is a PAH component, but the level of similarity with the comparison component (library instrument) does not reach the minimum standard of 85% [44,54,73]. Third, the components that appear are PAH derivatives. The PAH data from the four samples obtained from ST 1 (Table 4) indicate that the component is naphthalene (NL). The abundance of PAH components in the samples occurred in the following order: NL ˃˃ BZ ˃ AZ ˃ PR ˃ PH. The peak numbers in the recorded GS/MS chromatograms are sequential, according to the number of components [9,74]. However, some peak numbers are not included in the data (Table 1), indicating that the peak number is not considered to be a PAH component or that it may be a derivative of a type of PAH, but the level of similarity with the standard library on the instrument does not reach 85% [88,90,91].

The results of the identification of the PAH pollutant samples obtained from ST 2 were similar to those collected from ST 1. The PAH pollutant types, NL and PN, were found in all of the samples. The abundance of components in each sample was in the following order: NL ˃˃ PN ˃ PD. The AZ, BZ, and PH were only identified in the sediment samples, and PT was only found in the sponge samples (Table 5).

The results of the analysis of PAH pollutants obtained from ST 3 (Table 6) are similar to those for the samples from ST 1 and 2. The NL and PR were found in the four sample types, while the PT types were identified in the sediment, sponge, and water samples. AZ PAH pollutants were identified in the sediment and sponge samples, while BZ was only found in the sediment samples (Table 6) [54,79]. The abundance of components was, in order, NL ˃˃ PN ˃ PT. Comparing the variation of PAH components identified at all sampling stations (Table 4, Table 5 and Table 6), the types of PAH in the sediment samples were more varied, followed by the sponge samples, and then the seawater and fish samples [14,86,87]. The types of PAH pollutants identified in each sample showed that the PAH contained in the sediment samples were similar to those found in the sponge samples, while the seawater samples showed similar results to the fish samples [92,93]. This result is quite rational if we consider that sponges tend to settle at the bottom of the water (sediment), and the dominant habitat of fish is the water.

### 3.3. Pesticide Residue Pollutant Analysis and Medical Waste

It is suspected that pesticide residue and medical waste are contaminating the marine waters around BCI. This conclusion is based on the GC/MS chromatogram, which identified several types of organic compounds containing acidic groups (phenol, arsenous acid, sulfurous acid), carboxylic acid (hexadecenoic, benzoic acid), halogen groups (chloridanthene, fluoranthene, blomidanthene), phosphate, cyclopentasiloxane, thiophene, silicic acid, benzo[h]quinolone, and cyclotrisiloxane [16,40,42]. These compounds are thought to be derived from pesticide residues and medical waste and then change during their stay in the aquatic environment. This assumption was confirmed by the GC/MS chromatogram [77,94,95].

The components of pesticide residue and medical waste are not presented significantly in tabular form. The abundance of chromatogram peaks is due to the similarity or suitability of components with references in the tool library, which is less than 85% [12,26,96]. Determination of a component based on the GS/MS measurement is considered valid if it has a similarity level or quality of 80%, as was completed for the hydrocarbon and PAH components [14,17,42,48,97]. Ideally, for the analysis of pesticide residue and medical waste in samples obtained from marine ecosystems, such as from fish, sponge, seawater, and sediment, the preparation of these samples should be carried out using a combination of pyrolysis-GC/MS instruments [46,48,98,99].

### 3.4. Heavy Metal Pollutant Analysis

Many heavy metal pollutants are thought to enter the marine ecosystem. However, only Lead (Pb) and Copper (Cu) ions were identified in this research. Observations made in the Makassar City MTA area concluded that these two types of pollutants are the most significant components of pollution in marine areas, especially around BCI in the Spermonde Islands group [11,13,36,100].

Based on the quality standard according to KLH RI Num. 51/2004, the average concentration of the heavy metals Pb^+2^ and Cu^+2^ in seawater should be a maximum of 0.05 mg/L [4,10,33,62]. Comparing the concentrations identified in the results (Table 7) with the seawater quality standards showed that the pollutant concentrations of Pb^+2^ and Cu^+2^ around BCI were 20 times and 25 times higher than the required quality standards, respectively [44,48,62,100]. According to the Director General of BPOM RI No. 03725/B/SK/VII/89, the maximum threshold value of heavy metal contamination in fish for pollutants Pb^+2^ and Cu^+2^ = 0.008 µg/g [33,55,82,101]

The data on the average concentrations of the two heavy metal pollutants (Table 7) show that exposure to Pb^+2^ and Cu^+2^ is still below the threshold value. However, using the Australian and New Zealand Environment and Conservation Council (ANZECC) standard, with a smaller tolerance value than the maximum threshold, it was concluded that the two pollutants found in fish around BCI exceeded the standard [10,55,76,102,103]. Based on the data, Table 7 shows that all types of samples (fish, sponges, seawater, and sediments) obtained in the waters around BCI have exceeded the maximum threshold value specified [4,10,48,104]. This is understandable and quite rational because consumption patterns, activities, and dynamics in each type of marine fish vary greatly. Fish samples were obtained for analysis from different stations (Table 2). The type of fish sampled at ST 1 (Chrysiptera unimaculata) differed from the fish samples from ST 2 and ST 3 [62,105,106].

## 4. Discussion

Physical observation of the waters around BCI did not find plastic waste, media waste, or oil spills. The physical characteristics of Barrang Caddi waters (EC, TDS, pH, and temperature) and other parameters (Table 1) were similar to those of surrounding marine waters [107]. It is known that the contaminants in the GTP group are invisible. Demographically, the distance between BCI and Makassar City is ±11.5 km, the land area is ±45,000 m^2^, and the population is ±3000 people (612 families) (Figure 1). The residents generally make their livelihoods through the fishing industry, and some have vegetable gardens and undergo traditional household processing, generally burning [108,109]. Such conditions have ecological relationships and the potential to contribute to exposure to GTP contaminants [110].

The morphology of the fish and sponges differed at each station (Table 2). In marine ecosystems, fish interact with seawater as a growth medium throughout their lives, and the dominant sponges interact more with sediment (Figure 2, Figure 3, Figure 4 and Figure 5). These factors cause differences in life patterns, the dynamics that occur between types of fish and sponges, and differences in the nutritional patterns of the two types of marine biota (fish and sponges) in terms of their growth media, their level of preference for types of nutrients, and their limit of tolerance to GTP contaminants [111,112].

The results for the total abundance of MP for each type of sample analyzed (Figure 2 and Figure 3) showed that the waters around BCI are exposed to MP pollutants [18]. The type of MP contaminant contained in each sample type was analyzed further in terms of the density (film, fiber, fragment). The suspected types of MP based on polymer structures were distinguished by measurement and analysis, using the difference spectrum seen on the FTIR chromatogram (Figure 4 and Figure 5) [38,77]. These observations show that the marine ecosystem around BCI waters is exposed to almost all types of MP contaminants [34,113]. The dominant MP found in all samples was the mild MP group (PE, PP, PS), followed by the heavy group (PA and PC), while few of the PVC and ABS types were found (Table 3) [26,42]. The data illustrate that the dominant source of MP contaminants is community activities, including industry, services, and household activities [87]. This type of MP is widely used in packaging applications, such as in the manufacture of plastic bags, clothes, ropes, coloring, and recycling containers, as well as various types of plastic caps, bottles, disposable tableware, and optical discs (CDs). Almost all of the uses of this type of MP are part of the daily activities of society and industry [114,115].

The other GTP contaminants analyzed were hydrocarbon components, especially PAHs. Research was carried out on all types of samples from three different stations (Table 4, Table 5 and Table 6), and eight types of PAH were identified. In terms of types of low-toxic PAH, the NL type was identified in all samples. NL is a simple aromatic hydrocarbon [9,38,116]. Naphthalene-based products, such as camphor, are thought to come from household activities, for example, in products used to eradicate insects and absorb odors in bathroom toilets. Another source can be the distillation of coal tar. The nature of NL is volatile and carcinogenic [14,117].

Pyrene is a second order PAH that was identified in several samples. PN is a group of highly toxic and carcinogenic PAHs. The presence of PR in the sample is thought to have come from the transportation activities of petroleum carriers. This assumption was reinforced by the identification of the PAH type AZ, as the PAH type is a component of petroleum as are some of its derivative products, including those from coal processing [9,84,118]. This allegation is quite reasonable, because BCI operates three ports: a container port, a passenger port, and a fish-landing port. Other PAHs, such as PT, PH, and BZ, are derivative products resulting from chemical reactions, such as oxidation reactions that produce these derivatives or community waste burning activities [71,119].

Pesticide residues and medical waste are strongly suspected to be pollutants in the marine ecosystem around BCI. This assumption is based on GC/MS chromatogram peaks of varying abundances and the recorded GC/MS chromatogram peaks identified with PR and MW components and observed in the PAH chromatogram data (Table 4, Table 5 and Table 6), where the peak numbers not listed in the table are the peaks of the PR component or residual MW [73,109,116]. This assumption is based on the fact that two hospitals operate around the coast of Makassar City, which is adjacent to BCI [51,120]. The presence of the pollutant types PR and MW in samples of fish, seawater, sponges, and sediments around the marine waters of BCI could also have come from the activities of the people who inhabit BCI or be due to biological activities, where these pollutants undergo dynamic activities and eventually empty into the sea [24,55,96,110].

Heavy metal (HM) pollutants, especially Pb^+2^ and Cu^+2^, were identified in all of the samples obtained from the three different stations (Table 7). These results indicate that the concentrations are within the tolerable concentration limits for marine biota and are not dangerous according to Indonesian KLH standards, but they are in the worrying category based on ANZECC standards [12,62]. Pb^+2^ and Cu^+2^ pollutants in marine ecosystems around BCI are also thought to be due to natural and anthropogenic activities [20,121,122].

The concentration of all types of GTP identified in the waters around BCI is at alert status, not only in terms of the quality and sustainability of the marine ecosystem around BCI but also for the people who inhabit the island and the tourists visiting the island. In addition, the survival of sponges is important, as they are known to have primary and secondary metabolic contents that are used as raw materials for medicinal products, food concentrates, and beauty products [62,83,111,123].

It is estimated that exposure to contaminants from the GTP group around BCI will increase in the future due to the island’s status as part of the Marine Tourism Area (MTA), meaning that it is visited by many tourists, which is seen as a potential factor contributing to the pollutant component, especially the pollutant types MP and PAH [65,69,80]. BCI is adjacent to the city of Makassar, which has a beach length of ± 7.5 km. Anthropogenic sources of pollutants are thought to heavily contribute to this sector. Along the coast, there are hotels, hospitals, industrial operations, culinary areas, and residential housing [1,11,40]. The area is very densely populated, and community activities occur every day for almost 24 h per day [7,10,80]. This situation is thought to be the main factor causing exposure to MP pollutants of the MW type. The vicinity of BCI is also adjacent to the Soekarno-Hatta Port, a fish-landing port. Thus, there are many transportation activities for various types of ships. This significantly contributes to exposure to pollutants, especially pollutants of the PAH, HM, and MP types [13,46,101].

We recommend that the Makassar City MTA manager should use this data as material for consideration in MTA operations, so that the managed marine tourism does not cause increases in pollutants from the GTP category [3,47]. It is expected that the tourist area could be improved by increasing the population of marine biota, especially sponges, through the transplant method, as sponges are known as biota with functions, such as biodegradation, bio-absorption, and the bioremediation of several types of pollutants including PAHs, microplastics, and heavy metals [20,121]. Sponges can be used as bioindicators and for the biomonitoring of PAH and HM contamination [44,73,122]. Efforts should be made along the shoreline of BCI, and it is recommended that plants that have a biofilter function against pollutants, such as mangroves, should be planted [124]. Makassar City MTA managers are also advised to carry out socialization, education, and advocacy with the community to make them aware of the effects of waste, so that their household waste can be appropriately managed and not enter sea waters.

## 5. Conclusions

This article reviewed microplastic pollutants, PAHs, pesticide residue, heavy metals, and medical waste, which are known as global trending pollutants (GTP) and are present in marine ecosystems around BCI’s MTA waters. The average total MP abundance was found to be 5.47 units/m^3^ in seawater samples, 7.03 units/m^3^ in fish, 8.18 units/m^3^ in sediment samples, and 8.32 units/m^3^ in sponges. Seven types of MP were identified in the BCI aquatic ecosystem, dominated by the mild categories of MP, PE, PP, and PS, followed by PA and PC. The marine ecosystem around BCI was found to be contaminated with PAH-type pollutants, especially the NL type, which was found in all samples, followed by PN and AZ. BCI waters are also suspected of being exposed to pesticide residue and medical waste. The heavy metal pollutants Pb^+2^ and Cu^+2^ were also identified in the marine waters around BCI, especially in fish and seawater samples. The average concentrations of Pb^+2^ and Cu^+2^ in seawater were 0.164 ± 0.0001 mg/L and 0.293 ± 0.0007 mg/L, while the average concentrations of these two types of heavy metal pollutants were 1.811 ± 0.0002 µg/g and 4.372 ± 0.0003 µg/g, respectively.

The five types of pollutants identified in the marine ecosystem around BCI illustrate that this MTA is not conducive to being a recommended marine tourism destination. It is estimated that the concentrations of these pollutants will increase, given the status of BCI as one of the islands in the MTA area. If the BCI is maintained as a marine tourism area, it is feared that its ecosystem will enter an ecological hazard state, which will have a chain effect, not only on the aquatic ecosystem around the BCI but also on the sponge populations and fish, leaving them unfit for consumption and subsequently causing health problems for the community.

## Figures and Tables

**Figure 1 toxics-10-00301-f001:**
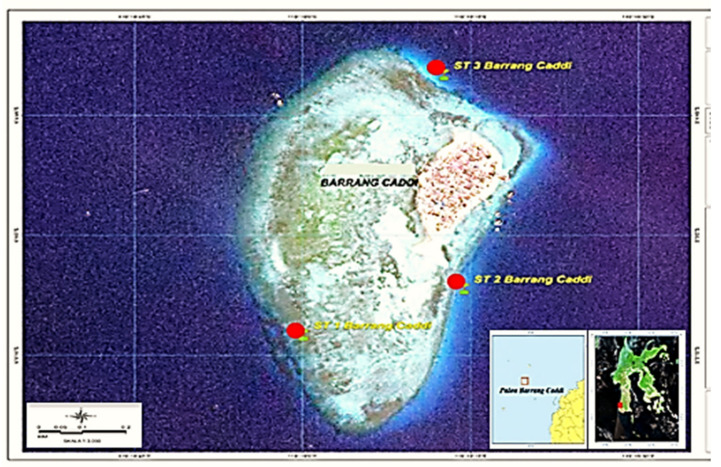
Map of Barrang Caddi Island (BCI). The red dots indicate the sampling stations (STs).

**Figure 2 toxics-10-00301-f002:**
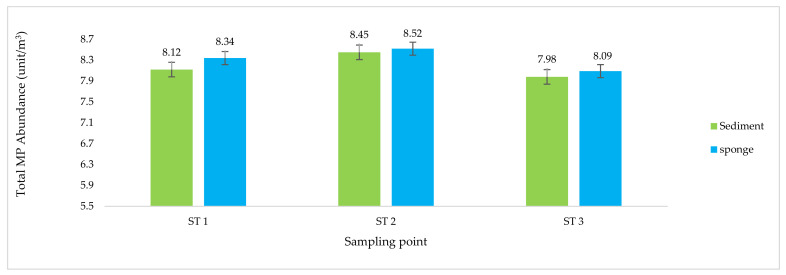
Total abundance of microplastics (MP) in the sediment and sponge samples based on the sampling points.

**Figure 3 toxics-10-00301-f003:**
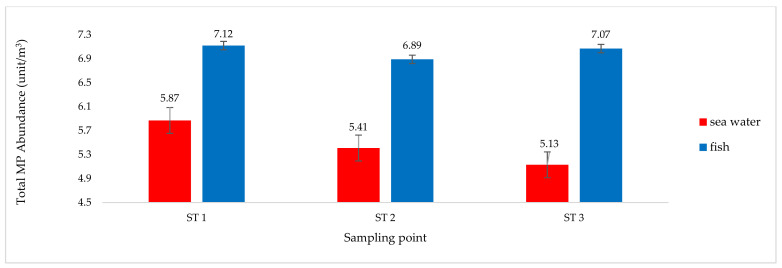
Total abundance of microplastics (MP) in seawater and fish samples based on the sampling points.

**Figure 4 toxics-10-00301-f004:**
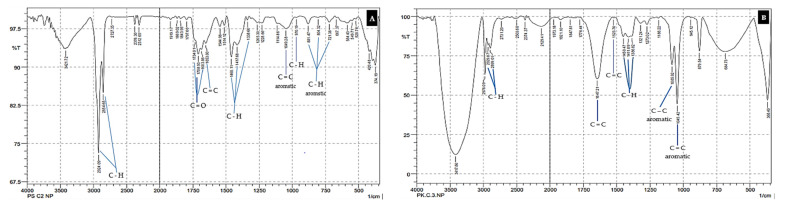
Identification of the type of MP pollutant in the sample based on the FTIR chromatogram. (**A**) Sediment samples; (**B**) sponge sample.

**Figure 5 toxics-10-00301-f005:**
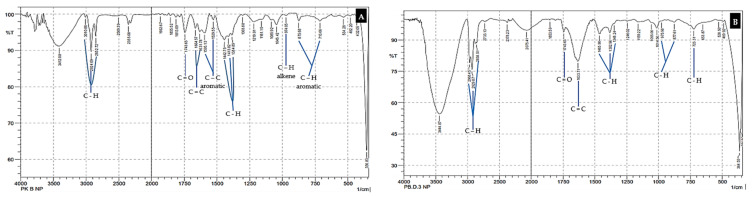
Identification of the type of MP pollutant in each sample based on the FTIR chromatogram. (**A**) Marine water samples; (**B**) fish samples.

**Table 1 toxics-10-00301-t001:** Physical characteristics of the seawater sampling stations in the waters around BCI.

Station Sampling	Coordinate	Depth MSL (m)	Distance from the Beach (m)	pH	Temperature (°C)	EC (ds/m)	TDS (mg/L)
ST 1	5°5′12.48216″ S 119°19′0.16536″ E	5	±300	7.32	29.8	14.91	7.56
ST 2	5°5′70.1664″ S 119°19′14.20716″ E	6	±350	7.30	30.4	16.69	8.12
ST 3	4°46′01664″ S 119°19′12.58932″ E	8	±370	7.31	29.9	16.33	8.22

**Table 2 toxics-10-00301-t002:** Fish and sponge morphology based on the sampling point.

Station Sampling	Sponges	Fishes
ST 1	*Cribrochalina olemda*	*Chrysiptera Unimaculata*
ST 2	*Clathria Reinwardtii*	*Ambhygphidodon Curacao*
ST 3	*Clathria* sp.	*Scolopsis Brenatus*

**Table 3 toxics-10-00301-t003:** Type and distribution of MP pollutants in each sample.

Station Sampling	Sample Type			
	Sediment	Sponges	Marine Water	Fishes
ST 1	PA, PC, PE, PP PS, ABS	PA, PE, PP, PS, ABS	PC, PE, PP, PS, PVC, ABS	PA, PE, PC, PP, PS
ST 2	PA, PC, PE, PP, PS	PA, PC, PE, PP, PS	PC, PE, PP, PS, ABS	PA, PC, PE, PP, PS
ST 3	PA, PC, PE, PP, PS, PVC	PA, PC, PE, PS, PVC	PA, PC, PP, PS, PVC	PA, PC, PE, PS, PVC

Note: PP = polypropylene; PE = polyethylene; PS = polystyrene; PC = polycarbonate; PA = polyamide nylon; PVC = polyvinyl chloride; ABS = acrylonitrile butadiene styrene.

**Table 4 toxics-10-00301-t004:** Types and abundance of PAH in samples collected from ST 1 based on the GC/MS chromatogram.

Sample Type	Peak Number	Retention Time	Peak Height	Quality (%)	Abundance (%)	Compound Name
Sediment	4	9.167	2,027,937	91	66,386	NL
	6	15.550	68,138	90	2.324	AZ
	9	17.788	26,924	85	0.786	PH
	12	19.023	31,894	87	1.278	PN
Sponge	2	9.168	2,565,156	91	76.539	NL
	6	19.023	28,386	93	2.410	PN
	7	26.736	62,698	87	11.672	BZ
Sea water	1	9.168	1,581,333	91	78.456	NL
	3	15.550	87,843	91	3.045	AZ
Fish	2	9.167	2,134,672	91	74.456	NL
	6	26.736	54,789	88	8.192	BZ

Note: the compounds included are those with a level of similarity (quality) reaching ≥ 85%. NL = Naphthalene group; AZ = Azulene class; PH = Phenyl group; PN = Pyrene group; BZ = Benzene group.

**Table 5 toxics-10-00301-t005:** Types and abundance of compound hydrocarbon and PAH in samples at ST 2 station based on GC/MS chromatogram.

Sample Type	Peak Number	Retention Time	Peak Height	Quality (%)	Abundance (%)	Compound Name
Sediment	1	9.168	1,125,323	91	84.221	NL
	2	13.205	20,184	96	1.186	PD
Sediment	3	15.549	54,212	85	3.643	AZ
	4	17.788	18,406	85	1.369	PH
	6	19.023	62,311	89	3.409	PN
	8	26.735	71,925	86	5.357	BZ
Sponge	3	9.167	3,219,575	91	86.046	NL
	4	13.206	44,603	97	0.978	PD
	6	16.283	14,355	86	0.696	PT
	8	19.023	31,674	87	1.312	PN
Sea water	3	9.168	860,498	91	63.945	NL
	4	13.205	20,327	96	1.204	PD
	9	19.023	138,767	96	10.165	PN
Fish	2	9.168	4,322,267	87	89.859	NL
	4	19.023	12,458	92	3.146	PN

Note: the compounds included are those that have a level of similarity (quality) reaching ≥ 85%. NL = naphthalene group; AZ = Azulene class; PT = phenanthrene; PN = Pyrene group; BZ = Benzene group; PH = Phenyl group; PD = Pentadecane.

**Table 6 toxics-10-00301-t006:** Types and abundance of PAH in samples obtained from ST 3 based on the GC/MS chromatogram.

Sample Type	Peak Number	Retention Time	Peak Height	Quality (%)	Abundance (%)	Compound Name
Sediment	2	9.168	1,222,751	91	82.356	NL
	4	15.549	88,566	90	3.087	AZ
	5	16.283	22,123	86	1.126	PT
	7	19.023	99,864	88	4.106	PN
	9	26.734	67,854	86	2.123	BZ
Sponge	2	9.167	2,423,789	91	78.127	NL
	4	15.550	65,832	85	3.983	AZ
	6	16.283	16,732	86	0.883	PT
	8	19.023	34,376	87	1.515	PN
Sea water	2	9.168	1,222,751	91	82.356	NL
	4	16.283	21,874	86	1.083	PT
	6	19.023	64,357	88	2.982	PN
Fish	3	9.168	1,111,244	91	64.362	NL
	5	19.023	13,318	87	0.734	PN

Note: the compounds included are those that have a level of similarity (quality) reaching ≥ 85%. NL = Naphthalene group; AZ = Azulene class; PT = phenanthrene; PN = Pyrene group; BZ = Benzene group.

**Table 7 toxics-10-00301-t007:** Heavy metal pollutant (HM) concentration analysis.

Type of Pollutant	Sampling Station	Average Concentration of Pollutants Replication (*n* = 3)			
		Sediment (µg/g)	Sponge (µg/g)	Sea Water (mg/L)	Fish (µg/kg)
Lead ion (Pb^+2^)	ST 1	4.041 ± 0.0004	3.871 ± 0.0003	0.104 ± 0.0002	2.452 ± 0.0003
	ST 2	4.676 ± 0.0003	3.725 ± 0.0002	0.251 ± 0.0002	1.656 ± 0.0001
	ST 3	4.643 ± 0.0005	3.813 ± 0.0004	0.137 ± 0.0005	1.326 ± 0.0004
Average:		4.453 ± 0.0003	3.803 ± 0.0003	0.164 ± 0.0001	1.811 ± 0.0002
Copper ion (Cu^+2^)	ST 1	9.279 ± 0.0001	6.166 ± 0.0002	0.319 ± 0.0002	4.822 ± 0.0001
	ST 2	8.843 ± 0.0001	5.567 ± 0.0003	0.286 ± 0.0003	4.474 ± 0.0002
	ST 3	7.920 ± 0.0002	5.474 ± 0.0002	0.275 ± 0.0004	3.821 ± 0.0004
Average:		8.681 ± 0.0004	5.735 ± 0.0007	0.293 ± 0.0007	4.372 ± 0.0003

## Data Availability

Data sharing is not applicable to this article.

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
