# Peer review of "Investigation of Global Trends of Pollutants in Marine Ecosystems around Barrang Caddi Island, Spermonde Archipelago Cluster: An Ecological Approach"

_toxics, 2022, doi:10.3390/toxics10060301_

Round 1
Reviewer 1 Report
General comments:
I would encourage the authors to work through the manuscript for consistency and clarity, particularly when describing where and how many sampling they take. How many replicas do they make in each determination? This is very important since It is not possible to obtain statistical data neither valid conclusions in a single study without replicas. Authors have to specify the number of water, sediment, sponges and fish samples. Please add a table.
Authors should give appropriate references or they must review the ones they have given because, in many sentences, there are not references that they can to support statements.
The methodology is confusing. This section must be modified, including a reproducible methodology or reference that clearly details.
ABSTRACT
Please, indicate the meaning of abbreviations before you use them for the first time (GTP, MP, PAH, PR, HM, MW, etc). Could be useful include a table at the beginning
Line 35-36: delete unnecessary “-“ This comment is general for the rest of the document.
Line 38: Is it really necessary to use so many decimal numbers? I my opinion so much accuracy is not possible. It is more appropriate to indicate the value of the standard deviation for each averaged.
- Materials and Methods
Please keep in mind that all devices and reagent used in this study should include information such as model, manufacturer, and country. Please, specify the concentration and the used reagent.
The methodology is confusing. This section must be modified, including a reproducible methodology or reference that clearly details it and the number of replicas (n)
2.2. Sampling
Please, indicate the quantity of samples/measurements (in total and per each sampling point and for each samples type (sponges, fish, sediment…).
Line 154: Please, indicate the meaning of TDS, HDL abbreviations before you use them for the first time.
Line 154: References of standard sampling methods should be added.
Line 168-170: Please, add reference
Equa. 1: Write m3 correctly, include superscript
Equa.2-4: In my opinion, this detail is unnecessary in a scientific article
Line 232: same as Equa. 1
Figure 2: Please, change “,” by “.” as decimal. The number of replicas must be indicate (n).
Figure 3: same as Figure 2
Application of proper quality assurance/quality control (QA/QC) procedures is vital for the measurement results to be treated as a source of reliable analytical information. Consequently, I suggest that a separate section devoted to QA/QC be added to the manuscript. Special attention should be paid to the description of the validation procedure for the applied analytical protocols, especially in the case of Cu and Pb measurements, elements present at the trace level
3.4. Heavy metal pollutant analysis
Applied methodology must be described properly. In addition, there is no information about the limit of detection (LOD) of determined elements. In this way. Why do you use AAS and no other with greater sensitivity, such as ICP-AES or ICP-MS?
Line 403: “only Lead (Pb) and Copper (Cu) ions were identified in this research” Explain why these elements (Pb and Cu) and not others (Cd, Zn, As…). Expand upon why and at what levels of exposure they are toxic, and why you would expect to find them in this environment. Better justification of the element choices is needed.
Line 406: Please, add reference
Line 412: https://doi.org/10.35932/ijcsrem.v2i1.16 the link is not correct
the reference can not to support statement. Please, include references to support this affirmation about seawater.
Line 414: the reference can not to support statement. Please, include references to support this affirmation about fish.
Table 7. Specify, for each row, the number of replicas (n) and the +/-
Author Response
Dear,
The manuscript has been revised according to the suggestion. Please find the Author's responses and revised manuscript attached to this mail. We are hopeful that this revision could be considered acceptable for publishing. Thank you.
Sincerely yours,
Author

Reviewer 2 Report
- The main object of the study in the abstract section should be rewritten again
- The abstract section contained many abbreviations, the abbreviation should be mentioned
- The design of the work is not clear and not organized in a scientific manner
- The method of the GC/MS chromatogram should be rewritten in details
- The whole manuscript needs English editing and updating the discussion section
Author Response

(The authors gave the same response as above.)

Round 2
Reviewer 1 Report
see file

Author Response
Dear reviewers
Thank you for providing a significant review for the improvement of our manuscript, so that the quality of this manuscript increases.
Author's response is attached in a word file
Thank you very much. It has helped a lot in improving the quality of our manuscript.
Makassar, 27 May 2022
Best regard
Authors

Reviewer 2 Report
The manuscript is now suitable for publication
Thank you
Author Response
Dear reviewers
Thank you for providing a significant review for improving our manuscript to increase the quality of this manuscript.
Thank you very much
Makassar, 27 May 2022
Best regards from us
Authors

This manuscript is a resubmission of an earlier submission. The following is a list of the peer review reports and author responses from that submission.